# Oral Tissue Involvement and Probable Factors in Post-COVID-19 Mucormycosis Patients: A Cross-Sectional Study

**DOI:** 10.3390/healthcare10050912

**Published:** 2022-05-13

**Authors:** Neelam Chandwani, Sandeep Dabhekar, Kalai Selvi, Roshan Noor Mohamed, Shahabe Saquib Abullais, Muhamood Moothedath, Ganesh Jadhav, Jaya Chandwani, Mohmed Isaqali Karobari, Ajinkya M. Pawar

**Affiliations:** 1Dentistry Department, All India Institute of Medical Sciences, Nagpur 441108, Maharashtra, India; drneelamch@gmail.com (N.C.); ganeshjadhav@gmail.com (G.J.); 2ENT Department, All India Institute of Medical Sciences, Nagpur 441108, Maharashtra, India; sandyent@aiimsnagpur.edu.in; 3Community Medicine, All India Institute of Medical Sciences, Nagpur 441108, Maharashtra, India; kalaiselvi@aiimsnagpur.edu.in; 4Department of Pediatric Dentistry, Faculty of Dentistry, Taif University, Taif 21944, Saudi Arabia; roshan.noor@tudent.edu.sa; 5Department of Periodontics and Community Dental Sciences, College of Dentistry, King Khalid University, Abha 61421, Saudi Arabia; drsaquib24@gmail.com; 6Department of Oral and Dental Health, College of Applied Health Sciences in Ar Rass, Qassim University, Buraydah 58876, Saudi Arabia; m.muhamood@qu.edu.sa; 7Department of Computer Science Engineering, Ramdeo Baba, College of Engineering and Management, Nagpur 440013, Maharashtra, India; jay13ch@gmail.com; 8Center for Transdisciplinary Research (CFTR), Saveetha Institute of Medical and Technical Sciences, Saveetha Dental College, Saveetha University, Chennai 600077, Tamil Nadu, India; 9Department of Restorative Dentistry & Endodontics, Faculty of Dentistry, University of Puthisastra, Phnom Penh 12211, Cambodia; 10Department of Conservative Dentistry and Endodontics, Nair Hospital Dental College, Mumbai 400008, Maharashtra, India

**Keywords:** COVID-19, corticosteroids, dentistry, diabetes mellitus, oral cavity, mucormycosis

## Abstract

The primary goal of this study was to assess the prevalence of oral involvement and, secondarily, the likely variables in patients with confirmed COVID-19 accompanied by mucormycosis infection. The study design was a cross-sectional descriptive sort that was performed at a tertiary centre. The non-probability convenience sampling approach was used to determine the sample size. Between May 2021 and July 2021, all patients who presented to our tertiary care centre with suspected mucormycosis were considered for the investigation. The research only included individuals with proven mucormycosis after COVID-19. The features of the patients, the frequency of intraoral signs/symptoms, and the possible variables were all noted. Of the 333 COVID-19-infected patients, 47 (14%) were diagnosed with confirmed mucormycosis. The mean (SD) age of the patients was 59.7 (11.9) years. Of the 47 patients with confirmed mucormycosis, 34% showed sudden tooth mobility, 34% expressed toothache, 8.5% reported palatal eschar, 34% presented with jaw pain, 8.5% had tongue discoloration, and 17% had temporomandibular pain. About 53% of the patients were known cases of type 2 diabetes mellitus, 89% of patients had a history of hospitalization due to COVID-19 infection, 89.3% underwent oxygen support therapy, and 89.3% were administered intravenous steroids during hospitalization due to COVID-19 infection. About 14% of the suspected cases attending the mucormycosis out-patient department (OPD) had been confirmed with definite mucormycosis. Oral involvement was seen in 45% of cases of CAM (COVID-associated mucormycosis). The most frequent oral symptoms presented in CAM were sudden tooth mobility and toothache. Diabetes and steroids were the likely contributing factors associated with CAM.

## 1. Introduction

With the inception of the COVID-19 pandemic in India since March 2020 and its rapid spread and upsurge during the second wave in 2021, the panic and havoc amongst Indians and rest of the world has risen [1,2]. The second wave of the COVID-19 pandemic showed its heavy impact, along with a complication in the form of the occurrence of COVID-19-associated mucormycosis in many patients across the world. However, its occurrence in India was huge compared to other parts of the world [1]. This presentation left the medical fraternity with a big challenge regarding the identification of its confirmatory cause, prevention, and management [1,2]. As most of the states in India, especially Maharashtra, have recorded a growing trend of confirmed mucormycosis cases in post-COVID-19 patients, the disease has been declared as an epidemic [3]. 

Mucormycosis (also called zygomycosis) is a rare, serious, and sometimes fatal opportunistic fungal infection caused by a group of moulds, called mucormycetes, that spreads rapidly. Rhizopus genera are considered to be amongst the principal causes of mucormycosis, followed by Mucor and Lichtheimia [4]. Mucorales are predominantly found in mud or soil, decaying food, dust, and manure [5]. Mucormycosis is caused by the inhalation of its filamentous (hyphal form) fungi, especially by immunosuppressed patients [6].

Mucormycosis enters into areas such as the sinuses, brain, lungs, and gastrointestinal tract. Depending on its site of occurrence, it can be classified into rhino-maxillary, rhino-orbito-maxillary with facial palsy, rhino-orbital cerebral mucormycosis, pulmonary, gastrointestinal, or disseminated [7,8,9]. It can even infect the oral cavity, skin, and other organs [10]. 

In the pre-COVID-19 period, its major manifestation in the oral cavity was palatal eschar, mandibular ulcer, and in rare instances, jaw necrosis [4,10,11]. Rhino-maxillary mucormycosis infection can cause oral involvement, with symptoms such as (1) maxillary sinus soreness, nasal obstruction, blackish or bloody discharge from the nose, face pain or numbness, and even abrupt tooth mobility [12,13,14]. This fungus is unique in that it begins its activities primarily by invading blood vessels (angiogenesis), culminating in thrombosis and tissue infarction [8]. As a result, swift diagnosis and treatment are required to avoid a high percentage of death and morbidity among suspected patients. The treatment recommendations for COVID-19 hospitalized patients during the pandemic included the use of corticosteroids in conjunction with supportive oxygen therapy, depending on the severity level [3]. Mucormycosis outbreaks in COVID-19 patients have been associated with irrational steroid use and the contamination of ventilators, oxygen concentrators, medical devices, and hospital disposables, such as linen, bandages, and so on [15]. Steroid use results in a compromised immune system, which makes COVID-19-positive patients more susceptible to opportunistic fungal infections. Steroids also have an established role in raising blood sugar levels, which is a big threat for patients with existing uncontrolled diabetes [13]. Most of the COVID-19 patients who had uncontrolled type 2 diabetes mellitus (DM) and were administered steroids for Severe Acute Respiratory Syndrome Coronavirus 2 (SARS-CoV-2) infection were more prone to mucormycosis infection [3,6,13]. Along with surgical debridement and medicinal management in the form of antifungal therapy, detecting the underlying associated risk factors for this infection is important for its comprehensive management to avoid a high percentage of morbidity and mortality amongst suspected patients [16]. Amongst the risk factors involved for mucormycosis, the most recorded risk factors for its occurrence are type 2 DM, neutropenia due to haematological complications, and the immunocompromised condition of the patients [17,18]. Despite the upsurge in the occurrence of cases of mucormycosis in post-COVID-19 patients, there is a deficiency in the data available due to the lack of studies based on population [19]. Hence, the current study aimed to assess the prevalence of oral involvement and, secondarily, the likely variables in patients with confirmed COVID-19 accompanied by mucormycosis infection.

## 2. Materials and Methods

The Institutional Ethical Committee (IEC/Pharmac/2021/280) approved the study. All individuals provided informed consent before being included in the study. The project’s aims, methods, and protocols were explained to the participants. The patients were told that the information gathered would be kept private.

The current study was a cross-sectional descriptive type of research conducted at a tertiary institution. The non-probability convenience sampling approach was used to determine the sample size. The research included all patients who presented to our tertiary care hospital with suspected mucormycosis between May 2021 and July 2021.

Our hospital had a designated screening area for post-COVID-19 mucormycosis. A total of 333 patients self-reported signs and symptoms of mucormycosis. The certificate of a positive RTPCR was validated. Inclusion criteria were any post-COVID-19 patients who had one or more of the following: 1. Facial discomfort, 2. Headache, 3. Nasal bleeding/discharge, 4. Sudden tooth movement, 5. Sudden toothache, 6. Pyloric ulcers, 7. Swelling of the face and orbits, 8. Loss of eyesight, 9. Haemoptysis. Amongst them, only patients with confirmed mucormycosis and COVID-19 confirmed by polymerase chain reaction test (either hospitalized or non-hospitalized) were included in the study. The mucormycosis was confirmed either by positive KOH or fungal culture of the nasal or by oral crust/swab and histopathology testing. 

Demographic characteristics, such as age and gender, were recorded. The frequency and percentage of intraoral signs/symptoms were assessed. 

The frequency of intraoral involvement was evaluated by examining the presence of six signs and symptoms as per the evidence available, namely, sudden tooth mobility, sudden toothache, jaw pain on the affected side, presence of palatal eschar or ulcer, tongue discoloration due to fungal infection, and temporomandibular joint pain. Each observation was given the mark 01. At least one 01 mark score out of six for each patient was considered as the criteria for having the presence of oral involvement in confirmed mucormycosis patients.

The presence of factors such as the presence of type 2 DM, a history of hospitalization, steroid administration, and oxygen administration during the COVID-19 period were recorded. 

## 3. Results

Following the screening protocol, 163 of the 333 post-COVID-19 patients who reported to the tertiary care hospital between May and July 2021 were suspected to have mucormycosis. Based on KOH and fungal culture tests, 47 patients (14.1%) were diagnosed with proven mucormycosis. There were 38 males (80%) and 09 females (20%) among the 47 patients. The patients’ mean (standard deviation) age was 59.7 (±11.9) years. The average period between the initial treatment for COVID-19 and the onset of mucormycosis was 15 to 45 days.

Merely 21 of the 47 individuals had oral signs and symptoms, whereas the remainder had no oral impression of the illness. All of the patients’ indications and symptoms were similar. More than two-thirds of individuals with oral involvement had more than two oral indicators of presentation. Out of 47 verified mucormycosis patients, 34% experienced abrupt tooth mobility, 34% indicated tooth soreness, and 8.5% reported palatal eschar (Figure 1). Moreover, 34% reported jaw discomfort, 8.5% reported tongue discoloration (Figure 2), and 17% reported temporomandibular joint pain.

At the time of hospitalisation for COVID-19, 25 of the 47 patients (53%) were recognised patients with type 2 diabetes (Table 1). The mean HbA1c value of the 47 patients was 6.95 (±1.76). Additionally, the HbA1c of the 21 individuals with oral symptoms was 7.14 (±1.81).

One hundred percent (Table 1) of the patients had a history of hospitalization due to COVID-19. The average length of stay in the hospital was 10.29 (±5.41) days. Oxygen support was provided to 89.3% of the patients (Table 1). Of them, 32 patients received oxygen by nasal prong at a rate of 2 to 5 L, 12 patients via high-flow nasal cannula at a rate of 10–15 L/min, and the remaining 03 were on non-invasive ventilator support. In addition, 42 patients (89.3%) were reported to have had intravenous steroid administration during hospitalization for COVID-19 for a minimum of three days and a maximum of twenty days.

## 4. Discussion

During the COVID-19 pandemic, it was discovered that SARS-CoV-2 infection, its treatment regimen, and the resulting decrease in immunity, as well as pre-existing comorbidities, rendered patients more prone to secondary infections, including mucormycosis. The symptoms that COVID-19-positive patients reported varied after being discharged from the hospital or recovering at home from the COVID-19 infection. It was even observed that some patients presented with the early indications of mucormycosis in the oral cavity, in the form of tooth soreness or tooth movement, which the dentist could easily overlook or disregard. Still, not much is known about the oral presentation in COVID-19-affected mucormycosis patients. Hence, it was important to understand the distribution of the intraoral presentation of this disease in post-COVID-19 patients so as to detect the disease as early as possible and prevent its rapid spread and mortality in affected patients.

The study assessed the patients’ gender and age characteristics, the frequency and percentage of intraoral signs and symptoms, and the associated factors of CAM patients. Male predominance of such patients was found with the percentage of 80%. This result is in accordance with previous literature [9], where the male predilection has been found in patients of mucormycosis. The mean age of the affected group was 59.7 (±11.91) years. This could be attributed to the fact that, in this age group, there are more chances for comorbidities [20].

This study was performed to investigate cases in which, during the second wave of the COVID-19 pandemic, patients were reporting to their dentists with vague oral symptoms, such as sudden toothache, without any prior history or without any carious lesion, sudden tooth mobility, or jaw pain. Moreover, as per past evidence [21], dental care for COVID-19 patients preceding such an infection, by means of a wound from post-tooth extraction or a post-tooth socket curettage wound, may make patients susceptible to fungal infections. Considering the fact that odontogenic symptoms may appear as the first sign and symptom [22,23,24,25,26] of this dreadful disease in post-COVID-19 patients, thorough evaluation, investigation, and early diagnosis are very important, especially in people with DM. Based on previous studies [23], it has been shown that, in patients without underlying immunosuppression, severe SARS-CoV-2-related pneumonia represents a low risk for invasive secondary fungal infection, especially aspergillosis. It can be presumed, based on the above evidence, that SARS-CoV-2 alone may have the least possibility of being a risk factor for mucormycosis.

As most of the patients had combined nasal and sinus symptoms along with oral symptoms, they may have presented with sudden toothache and sudden tooth mobility in the absence of any previous carious lesion. This may be due to the encroachment of infection from the maxillary sinus or nasal cavity towards the tooth root on the side of sinus involvement. Palatal eschar was noted in only 04 patients with palatal bone erosion. Palatal eschar has previously been noted in patients with the clear involvement of the palatal bone perforating from the maxillary side toward the oral cavity [24]. Referred jaw pain due to the spread of infection in the maxillary sinus was observed in 16 patients, which may be due to the involvement and spread of infection in the maxillary sinus and nasal cavity in most of the patients. The whitish tongue discoloration was presented in only 04 patients, which may be due to the candidiasis infection, either because of steroid consumption or poor oral hygiene [25,26].

No such studies have been done in the past to evaluate the distribution of oral signs and symptoms in COVID-19-associated mucormycosis patients.

In the present study, it was found that 25 (53%) patients had associated type 2 DM. The literature provides evidence that DM is a relevant risk factor for the development of rhino-cerebral mucormycosis [7,12].

Out of 47 patients, 42 (89.3%) patients were hospitalized due to COVID-19, and all of them had a history of steroid consumption and oxygen therapy during hospitalization. The average duration of hospitalization in all 47 patients was 10.29 days with standard deviation 5.41. Steroids are known to suppress immunity, which makes COVID-19-positive patients more prone to opportunistic fungal infections. Steroids also have an established role in raising blood sugar levels, which is a big threat for COVID-19-positive patients with existing uncontrolled diabetes [18,27].

Steroid consumption, hospitalization, and oxygen therapy, along with the DM, could be considered as the probable risk factors for the development of rhino-maxillary mucormycosis. Still, further research is needed considering the limited sample size and limited parameters included in the study.

As this was a cross-sectional study, the findings were noted on the first day of admission, and there was no subsequent follow-up taken. However, all these patients were admitted and given the medical and surgical treatment as per need in the hospital. The availability of Amphotericin, which is the first drug of choice, was easily available in the hospital. Patients having only sinus involvement were managed by functional endoscopic sinus surgery. Patients having palatal involvement were managed by medial maxillectomy and total maxillectomy as per the case.

## 5. Conclusions

About 14% of the patients attending the mucormycosis OPD had been confirmed with definite mucormycosis. Oral involvement was seen in 45% of the cases of CAM. The most frequent oral symptoms presented in CAM were abrupt tooth mobility and toothache. Diabetes and steroids were the probable contributing factors associated with CAM.

It could be concluded that oral or tooth-related signs and symptoms in post-COVID-19 individuals should not be neglected or overlooked. Thorough history and investigations should be initiated as soon as possible to rule out deadly mucormycosis infection so as to prevent morbidity and mortality in patients.

## Figures and Tables

**Figure 1 healthcare-10-00912-f001:**
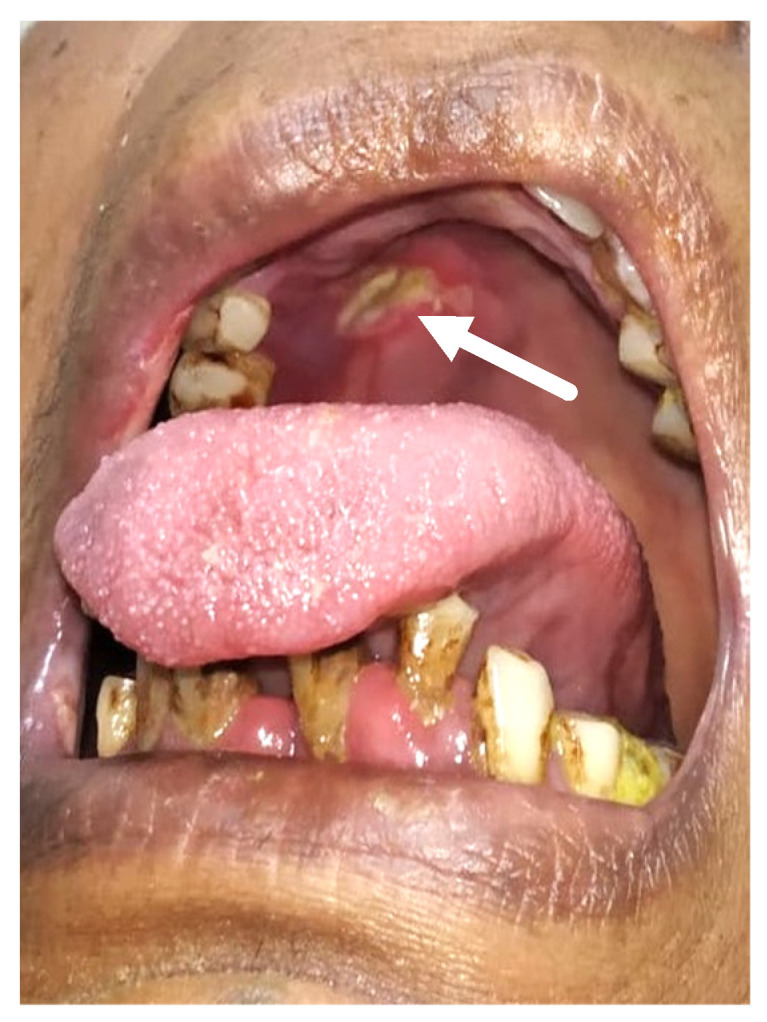
This is a clinical picture showing palatal eschar (white arrow).

**Figure 2 healthcare-10-00912-f002:**
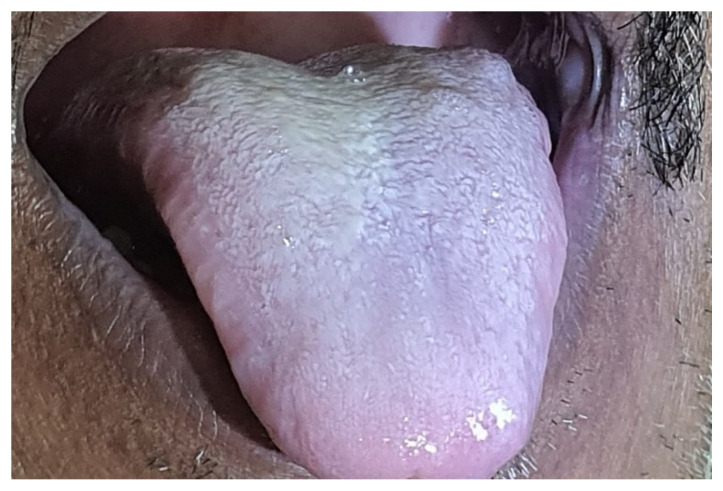
This is a clinical picture showing a discoloured tongue of a patient with confirmed mucormycosis.

**Table 1 healthcare-10-00912-t001:** Clinical characteristics of patients with confirmed mucormycosis.

Factors	Frequency	Percentage (%)
Diabetes Mellitus (DM)	25	53.1
History of Hospitalization	47	100
Oxygen Therapy	42	89.3
Steroid Consumption	42	89.3

## Data Availability

The data set used in the current study will be made available at reasonable request.

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
