# Peer review of "Oral Tissue Involvement and Probable Factors in Post-COVID-19 Mucormycosis Patients: A Cross-Sectional Study"

_healthcare, 2022, doi:10.3390/healthcare10050912_

Round 1
Reviewer 1 Report
congratulations on the manuscript, I consider it important because it evidences a clinical manifestation in the spectrum of mucormycosis that is useful to take into account for the clinician in his evaluation, it contributes in the scenario of the COVID 19 pandemic, the number of cases is quite representative as this is a previously rarely reported disease. It is very interesting that the manifestations of the oral cavity were reported in almost 50% of the confirmed cases, with dental pain and dental mobility being the most representative that could be overlooked in the evaluation if an adequate investigation of the patient is not conducted.Author Response
Point 1: Congratulations on the manuscript, I consider it important because it evidences a clinical manifestation in the spectrum of mucormycosis that is useful to take into account for the clinician in his evaluation, it contributes in the scenario of the COVID 19 pandemic, the number of cases is quite representative as this is a previously rarely reported disease. It is very interesting that the manifestations of the oral cavity were reported in almost 50% of the confirmed cases, with dental pain and dental mobility being the most representative that could be overlooked in the evaluation if an adequate investigation of the patient is not conducted.
Response 1: Thank you so much respected reviewer for the encourging comments.
Reviewer 2 Report
Please find my comment for the manuscript ID healthcare-1651373 titled "Oral tissue involvement and probable factors in post Covid-19 mucormycosis patients: A cross sectional study".
- Authors have presented the prevalence of oral involvement variables in post-recovery of Covid-19 pt accompanied by mucormycosis but there was information about either the covid-19 infections reported in vaccinated or unvaccinated pt.
- Line 111, are you sure about this , "Total 333 patients reported themselves as post Covid-19 Mucormycosis suspects", what was the inclussion criteria?
- It would be interesting if you would had compared the signs and symptoms of sudden tooth mobility, sudden tooth ache, jaw pain, eschar or ulcer, tongue discoloration and temporomandibular joint pain in both suspected and confirmed cased of mucormycosis.
- Mucormycosis confirmed pts were 47/333 patients (14.1%), and only 21 of 47 pts had oral signs and symptoms. How about the remaining suspected mucormycosis pt, were there any Covid-19 related oral manifaction as mentioned in your criteria? Did you took the duration of post Covid-19 infection in account?
- Line 132-145 have a repeatation of results.
- Results section in very weak, please re-write and descibe your results systamatically.
- Line 117-119 "Assessment of Patients ..... presence of type 2 DM, history of hospitalization, steroid administration......" should be re-written and these finding should be presented in the Result and Discussion sections in a better way
- Line 234, "oral involvement was seen in 45%....", looks untrue as percentages presented and data description require more care interpration.
- Authors are encouraged to cite and review the following papers:
- Janjua OS et al. Dental and Oral Manifestations of COVID-19 Related Mucormycosis: Diagnoses, Management Strategies and Outcomes. J Fungi (Basel). 2021 Dec 31;8(1):44. doi: 10.3390/jof8010044.
- Al-Tawfiq J et al. COVID-19 and mucormycosis superinfection: the perfect storm. Infection. 2021 Oct;49(5):833-853. doi: 10.1007/s15010-021-01670-1.
Author Response
Point 1: Authors have presented the prevalence of oral involvement variables in post-recovery of Covid-19 pt accompanied by mucormycosis but there was information about either the covid-19 infections reported in vaccinated or unvaccinated pt.
Response 1: Thank you for bringing this to our attention, respected reviewer. The vaccination status of the patients was not taken into account in the current investigation since the majority of the patients were unvaccinated or just partially vaccinated because immunisation was not yet widely implemented in India. During that time, only healthcare professionals were vaccinated on a priority basis.
Point 2: Line 111, are you sure about this , "Total 333 patients reported themselves as post Covid-19 Mucormycosis suspects", what was the inclussion criteria?
Response 2: Thank you for bringing this to our notice, respected reviewer. We have added now in the manuscript [Lines 115 - 120] “A total of 333 patients self-reported signs and symptoms of mucormycosis. The certificate of RTPCR positive was validated. Inclusion criteria were any post-covid patients who had one or more of the following: 1. Facial discomfort, 2. Headache, 3. Nasal bleeding/discharge, 4. Sudden tooth movement, and 5. Sudden toothache 6. pyloric ulcers, 7. Swelling of the face and orbits, 8. loss of eyesight, and 9. Hemoptysis”
Point 3: It would be interesting if you would had compared the signs and symptoms of sudden tooth mobility, sudden tooth ache, jaw pain, eschar or ulcer, tongue discoloration and temporomandibular joint pain in both suspected and confirmed cased of mucormycosis.
Response 3: Thank you for your useful feedback, respected reviewer. We would want to inform the esteemed reviewer that this was not the key objective of the research design and we did not compare mucormycosis and non-mucormycosis patients. The described cross-sectional investigation solely included Covid related mucormycosis patients.
Point 4: Mucormycosis confirmed pts were 47/333 patients (14.1%), and only 21 of 47 pts had oral signs and symptoms. How about the remaining suspected mucormycosis pt, were there any Covid-19 related oral manifaction as mentioned in your criteria? Did you took the duration of post Covid-19 infection in account?
Response 4: Out of 163 CAM suspects, 27 showed one or more oral signs and symptoms, 21 of which were proven mucormycosis and the remaining 06 were not associated to mucormycosis since they tested negative on the KOH / fungal culture test. The presentation in these 06 patients were as follows 02 patients had grade I tooth mobility due to poor periodontal condition. 04 had tooth ache due to obvious dental caries.
When screening for CAM, the period of post-Covid-19 infection was not taken into account. Any post-Covid patient, regardless of duration, was considered a suspect.
Point 5: Line 132-145 have a repeatation of results.
Response 5: We apologise for this blunder and we have ammended the concern and deleted the repeted lines.
Point 6: Results section in very weak, please re-write and descibe your results systamatically.
Response 6: We apologise for this and we have ammended the concern and changed the results section. Hopefully now acceptable.
Point 7: Line 117-119 "Assessment of Patients ..... presence of type 2 DM, history of hospitalization, steroid administration......" should be re-written and these finding should be presented in the Result and Discussion sections in a better way
Response 7: We apologise for this and we have ammended the concern and changed the results section. Hopefully now acceptable.
Point 8: Line 234, "oral involvement was seen in 45%....", looks untrue as percentages presented and data description require more care interpration.
Response 8: We have ammended the concern and changed conclusion with simplified explanation [Lines 247-249].
Point 9: Authors are encouraged to cite and review the following papers:
Janjua OS et al. Dental and Oral Manifestations of COVID-19 Related Mucormycosis: Diagnoses, Management Strategies and Outcomes. J Fungi (Basel). 2021 Dec 31;8(1):44. doi: 10.3390/jof8010044.
Al-Tawfiq J et al. COVID-19 and mucormycosis superinfection: the perfect storm. Infection. 2021 Oct;49(5):833-853. doi: 10.1007/s15010-021-01670-1.
Response 9: We have ammended the suggestion and added the said references (Ref No. 26 and 27).
Reviewer 3 Report
Comments for Manuscript ID: healthcare-1651373
Authors reported clinical characteristics of covid-19-related mucormycosis with stomatognathic diseases. The report includes important information for health professions, especially dentist. To improve readability of the manuscript, several minor comments are shown below.
Abstract
Lines 42-46 These sentences overlap with the sentences in lines 35-42.
Line 43 Please indicate “OPD” in full spelling.
Introduction
Lines 52-55 These statements need references.
Results
Lines 139-143 These sentences overlap with the sentences in lines 132-136.
Please change the word “Hba1c“ to “HbA1c”
As variables, nutritional condition or socioeconomic status of patients will strengthen the report. Recently, there are increasing reports of breakthrough infections of mucormycosis during voriconazole administration. Are there patients administrated voriconazole in clinical history?
To diagnose of periodontitis, were patients photographed X-ray? If so, please kindly report status of alveolar bone in patients.
What is the average period between initial treatment for covid-19 and onset of mucormycosis?
Author Response
Point 1: Abstract
Lines 42-46 These sentences overlap with the sentences in lines 35-42.
Response 1: Thank you for bringing this to our attention, respected reviewer. We have ammended this comment in abstract.
Point 2: Line 43 Please indicate “OPD” in full spelling.
Response 2: Thank you for bringing this to our notice, respected reviewer. We have ammended this comment [Line 45].
Point 3: Lines 52-55 These statements need references.
Response 3: We have now added the reference [Line 59].
Point 4: Lines 139-143 These sentences overlap with the sentences in lines 132-136.
Response 4: We have changed the results section and hopefully now acceptable.
Point 5: Please change the word “Hba1c“ to “HbA1c”
Response 5: We apologise for this blunder and we have ammended the concern [Lines 164 and 165].
Point 6: As variables, nutritional condition or socioeconomic status of patients will strengthen the report. Recently, there are increasing reports of breakthrough infections of mucormycosis during voriconazole administration. Are there patients administrated voriconazole in clinical history?
Response 6: There was no such reported history of use of Voriconazole in any of the patients.
Point 7: diagnose of periodontitis, were patients photographed X-ray? If so, please kindly report status of alveolar bone in patients.
Response 7: A Ct scan of the patients if advised was done however no such advise was given to the patients.
Point 8: What is the average period between initial treatment for covid-19 and onset of mucormycosis?
Response 8: The average period between initial treatment for covid-19 and onset of mucormycosis was 15 to 45 days.
Reviewer 4 Report
Oral tissue involvement and probable factors in post Covid-19 mucormycosis patients: A cross sectional study
Chandwani et al reported a description of 47 cases Covid-19 associated mucormycosis (CAM). It is a interesting topic, but unfortunately the manuscript need significant improvements/changes.
Here some comments:
- There is no connection between title, goals, results, and conclusions. The title generates the idea of a more complex analysis aimed to identified ¨probable factors¨ but the study is only limited to a description of CAM cases (it is not adding novel information). It is important to said that descriptive studies are relevant, in special in CAM, but in this case, authors did not achieve the proposed objectives. Maybe a cases series description could be a better option.
- The results section is lacking of information. I expected to find more detailed data about the description of how CAM diagnose was done, like, number of patients with positive microscopy and culture, isolates identification (those with positive culture), description of treatment (antifungal and surgical), outcomes (survival and sequelae in survivors).
- Results section need extensive rewriting, there is not a flow. In addition, there are some statements duplicated (page 2, line 78, page 4, line 139).
- Please check writing style, including terminology, and format of presentation of data (total numbers and percentages).
- Authors are presenting results in the discussion, an example in page 5, line
- I would like to suggest reconsider the use of the term ¨few¨ in the first paragraph of the discussion (page 4, line 168). About 50% of cases with oral lesion is not few.
Author Response
Point 1: Chandwani et al reported a description of 47 cases Covid-19 associated mucormycosis (CAM). It is a interesting topic, but unfortunately the manuscript need significant improvements / changes.
Response 1: Thank you for the encouraging comments.
Point 2: There is no connection between title, goals, results, and conclusions. The title generates the idea of a more complex analysis aimed to identified ¨probable factors¨ but the study is only limited to a description of CAM cases (it is not adding novel information). It is important to said that descriptive studies are relevant, in special in CAM, but in this case, authors did not achieve the proposed objectives. Maybe a cases series description could be a better option.
Response 2: We have mad changes in the Abstact aswell as the Manuscript to make it appealing to the reviwer and readers.
Point 3: The results section is lacking of information. I expected to find more detailed data about the description of how CAM diagnose was done, like, number of patients with positive microscopy and culture, isolates identification (those with positive culture), description of treatment (antifungal and surgical), outcomes (survival and sequelae in survivors).
Response 3: Thank you respect reviewer for a valuable insight. All suspects underwent the KOH test/ fungal culture of nasal or oral crust / secretion. Out of those 47 patients were found to be positive for mucormycosis. As this was a cross sectional study the patients were not followed till the outcome.
Point 4: Results section need extensive rewriting, there is not a flow. In addition, there are some statements duplicated (page 2, line 78, page 4, line 139).
Response 4: We have changed the results section and hopefully now acceptable.
Point 5: Authors are presenting results in the discussion, an example in page 5, line
Response 5: We apologise for this blunder and we have ammended the concern [Lines 191 to 196].
Point 6: I would like to suggest reconsider the use of the term ¨few¨ in the first paragraph of the discussion (page 4, line 168). About 50% of cases with oral lesion is not few.
Response 6: We apologise for this blunder and we have ammended the concern [Lines 185].
Round 2
Reviewer 2 Report
Thank you, the authors have provided replies to my comment. But I feel this manuscript should be thoroughly proofread.
- Line 163-165 is not clear, please rewrite.
- Line 164-166, HbA1c values of 6.95 (±1.76) vs 7.14 166 (±1.81) is statistically significant?
- Table 1, should also include the values of data with the number of patients
- Line 229, the history of steroid consumption was not mentioned is not clear, as in the results section you mentioned IV steroid administration for 3-20 days. Please comment about any possible association between mucormycosis and steroids.
- Please correct the highlighted area and carefully re-write.
